# Development of novel monoclonal antibodies for detection of *pan*-Lassa virus

Ruchi Paroha[1], Takeshi Saito[1], Shintaro Yamada[1], Christine Click[2], Slobodan Paessler[1], Junki Maruyama [1,3]*

1 Department of Pathology, The University of Texas Medical Branch, Galveston, Texas, United States of America, 2 Department of Biochemistry and Molecular Biology, The University of Texas Medical Branch, Galveston, Texas, United States of America, 3 Institute for Human Infections and Immunity, The University of Texas Medical Branch, Galveston, Texas, United States of America

* jumaruya@utmb.edu

## Abstract

Lassa virus (LASV), the causative agent of Lassa fever (LF), poses a significant public health concern in endemic regions due to its high morbidity and mortality. Rapid and accurate diagnosis is critical for effective clinical management and containment during LF outbreaks. However, the genetic diversity of LASV, encompassing at least seven distinct lineages, poses a major challenge for the development of broadly reactive diagnostic tools. Therefore, there is an urgent need to develop detection methods that are effective across diverse lineages. To address this challenge, we generated a panel of cross-reactive monoclonal antibodies (mAbs) targeting the LASV nucleoprotein (NP). Several mAbs exhibited broad reactivity across LASV lineages I-VII in different immunoassays such as enzyme-linked immunosorbent assay (ELISA), western blotting, and immunofluorescence assay. Utilizing these broadly reactive mAbs, we developed an antigen-capture sandwich ELISA capable of detecting LASV NP from all seven lineages with high sensitivity. Our findings highlight a set of novel mAbs with broad cross-reactivity across lineage. In addition, we demonstrated their utility in a sandwich ELISA format for *pan*-LASV detection. This *pan*-LASV diagnostic approach offers a promising tool for improved LF diagnosis in both clinical and epidemiological settings.

## Author summary

This study addresses the urgent need for improved diagnostic methods for LF. Different lineages of LASV complicates the development of broadly effective detection tools, making it challenging to design a single test capable of detecting all the lineages. To overcome this challenge, we generated mAbs that recognize the NP of LASV across all currently known lineages. These antibodies were used to develop sandwich ELISA, enabling the capture and effective identification of LASV. The findings suggest that these novel mAbs hold significant promise for

**Data availability statement:** All data required to replicate the findings (raw data) are deposited at figshare (https://doi.org/10.6084/m9.figshare.31424906).

**Funding:** This work was supported by National Institute of Health K99AI156012, R00AI156012 (JM), U01AI151801 (SP), The Institute for Human Infections and Immunity (JM), and John S Dunn Foundation (SP). The funders had no role in study design, data collection and analysis, decision to publish, or preparation of the manuscript.

**Competing interests:** The authors have declared that no competing interests exist.

developing a LASV diagnostic assay suitable for all known lineages of the virus. This could lead to earlier diagnosis and better management of the disease surveillance in endemic regions.

## Introduction

LF, caused by LASV, is endemic in several west African countries, which leads to a significant public health threat in endemic areas. Globally, it is estimated that approximately 300,000–500,000 infections and about 5,000 deaths occur each year due to lack of efficacious countermeasures [1]. Since 2015, the number of cases has been increasing and LF poses significant global public health concerns [2]. LASV belongs to genus *Mammarenavirus*, family *Arenaviridae*, and its genome consists of bi-segmented, single stranded, negative-sense RNA. The large (L)-segment encodes the RNA-dependent RNA polymerase, and the multifunctional matrix protein (Z), which plays an important role in viral assembly and budding. The small (S)-segment encodes the glycoprotein precursor (GPC), which is post-translationally cleaved into stable signal peptide (SSP), GP1, and GP2 for facilitating viral entry into the host cell, and the NP, which is essential for viral replication and immune evasion [3]. In spite of high distribution and high mortality rate, vaccine and other therapeutic measures are very limited for LF. Several vaccine candidates based on recombinant vesicular stomatitis virus (VSV), measles, rabies and DNA vaccine under clinical trials which have demonstrated promising efficacy in animal models [4,5] but, till date, no vaccine has been approved for LF. Standard treatment against LF relies on the off-label use of ribavirin, nonspecific nucleoside analog [6,7], in combination with supportive cares in the endemic area. However, ribavirin needs to be administrated early, ideally within a week of onset of symptom to show better efficacy [8–10]. Due to the unavailability of vaccines and therapeutic measures, early diagnosis of LF is a key component in implementing effective patient care and containment strategies for LF outbreaks.

Current diagnostic methods for LF are often applied after medical examination for the presentation of clinical symptoms, such as fever, headache, cough along with history of exposure or travel in endemic area. Furthermore, co-infection with other pathogens such as malaria and HIV complicates clinical diagnosis, making symptom-based identification unreliable [11]. To address these challenges, various laboratory-based diagnostic techniques have previously been developed for LASV detection. Virus isolation and identification are the 'gold standard' for LASV detection [12]. However, this method is time-consuming and requires specialized biosafety precautions, as LASV belongs to risk group 4 pathogens and must be handled in high-containment facilities. For further virus characterization, such as strain identification, other molecular and serological assays need to be performed in addition to virus isolation. Molecular diagnostics, e.g., reverse transcription polymerase chain reaction (RT-PCR)-based virus identification, are currently the preferred methods for LF diagnosis and have been routinely employed for clinical and laboratory-based LASV diagnosis [13]. Although this method offers good specificity, high sensitivity, and

rapidity, its application in rural areas is significantly limited. It requires specialized training of personnel, access to expensive instruments and the necessary molecular reagents, which are often limited in rural settings in endemic areas, thereby restricting the use of PCR-based methods for routine testing. Other than technical challenges, RT-PCR based assays also lack lineages wide LASV detection due to their genetic diversity [14,15]. Antigen-antibody reaction-based detection methods are another option. Previously developed antigen-capture ELISAs could either detect region-specific LASV lineage or could only detect lineage II, III and IV, which are considered as common lineages in the endemic area [10,16,17]. Additionally, there are concerns about the accuracy of some diagnostics from the endemic area [18]. Given the fact that LASV lineages are not region specific and may overlap in some cases, i.e., multiple lineage infections may occur in one single region, and additionally such lineage-specific tests are not applicable for diagnosis of imported cases worldwide [19].

Therefore, despite the range of tests available, a diagnostic gap remains, underscoring the need for a simple, specific, and reliable pan-LASV detection method for LF diagnosis as a. Any LASV detection assay covering the diversity of LASV genetic variation across West Africa is not available up to now. In this study, we have successfully generated anti-LASV NP mAbs with cross-reactivity for LASV lineage I-VII. Additionally, we have developed an antigen-capture sandwich ELISA for *pan*-LASV detection using novel mAbs.

## Materials and methods

### Ethics statement

All animal studies were reviewed and approved by the Institutional Animal Care and Use Committee at The University of Texas Medical Branch, Galveston (IACUC# 2003029) and were conducted according to the National Institutes of Health guidelines.

### Cells and viruses

African green monkey kidney Vero cells and human embryonic kidney HEK293T cells were grown and maintained in Dulbecco's modified Eagle's medium (DMEM, Sigma-Aldrich) supplemented with 10% fetal bovine serum (FBS, Gibco) and 1% penicillin-streptomycin (Gibco). Mouse myeloma P3U1 cells were cultured in Roswell Park Memorial Institute (RPMI) 1640 medium (Thermo Fisher Scientific) supplemented with 10% FBS and 1% penicillin-streptomycin. Cells were maintained in a humidified incubator at 37°C with 5% $CO_2$. ML29, a reassortant of Mopeia virus and LASV [20] was kindly provided by Dr. Igor S. Lukashevich (University of Louisville, Louisville, KY, USA) and propagated in Vero cells. Recombinant vesicular stomatitis virus (rVSV) was rescued as described previously [21] and propagated in Vero cells. LASV strain Pinneo (lineage I), Sauerwald (lineage II), Ojoko (lineage III) were obtained from Dr Thomas G. Ksiazek, virus collection at The University of Texas Medical Branch, Galveston (UTMB). LASV LF2384 (lineage IV) was isolated from a fatal LF case during a 2012 outbreak in Sierra Leone as previously described [22,23] and were propagated in Vero cells. Virus titer was determined by plaque assay and represented as plaque-forming units (PFU) as described previously [22]. Ebola virus Zaire 199510621 (EBOV) and Marburg virus Angola 200501379 (MARV) were obtained from Dr Thomas G. Ksiazek, virus collection at UTMB. EBOV and MARV were propagated in Vero E6. Lymphocytic choriomeningitis virus (LCMV) stain Armstrong was kindly provided by Dr. Juan C de la Torre, The Scripps Research Institute, CA, USA. All works with infectious LASVs, EBOV, and MARV were performed in biosafety level 4 (BSL-4) laboratory in the Galveston National Laboratory (GNL), at UTMB according to the institutional safety guidelines and the Federal Select Agent Program guidelines.

### Plasmids construction

pCAGGS plasmids expressing LASV NP (pC-LASV-NP), GPC (pC-LASV-GPC), and Z (pC-LASV-Z) were constructed in our laboratory. Briefly open reading frame (ORF) of NP of LASV strains Pinneo, Sauerwald, Ojoko, and LF2384 were amplified using KOD One PCR Mastermix (Toyobo) and cloned into pCAGGS vectors using In-Fusion HD cloning kit

(Clontech) according to the manufacturer's instructions. The ORF of NP, GPC and Z of LASV strains Soromba-R (lineage V, accession numbers, S segment; KF478765.1, L segment; KF478762.1), LASV/H.sapiens-tc/NGA/2016/IRR_006 (IRR_006, lineage VI, accession numbers, S segment; MK107927.1, L segment; MK107872.1) and, Togo/2016/7082 (lineage VII, accession numbers, S segment: KU961971.1, L segment: KU961972.1) were synthesized by Twist Bioscience and cloned into pCAGGS vectors using In-Fusion HD cloning kit. All the constructed plasmids were confirmed by Sanger sequencing before using in the study.

## Preparation of virus-like particles (VLPs)

HEK293T cells were transfected with equal amounts of pC-LASV-NP, -GPC and -Z using FuGENE HD transfection reagent (Promega) according to the manufacturer's instructions. At 48 hours post-transfection, cell culture supernatant was collected and centrifuged 3,000 rpm for 10 mins to remove cell debris and filtered using 0.45 µm filter (Merck Millipore). VLPs were purified from cleared culture supernatant using 20% sucrose cushion with ultracentrifugation for 2 hours at 28,000 x g, 4°C. The VLPs were resuspended in phosphate-buffered saline (PBS). The expressions of NP, GPC, and Z were confirmed by western blotting (WB) as described below. Since anti-Z antibody could not be able to detect LASV IRR_006 Z, pC-LF2384-Z was used for IRR_006 VLPs preparation along with pC-LASV IRR_006-NP and-GPC.

## Preparation of LASV NPs as antigens

LASV NP from seven lineages were prepared as antigens according to the previously described study [24]. Briefly, HEK293T cells were transfected with 1 µg of pC- LASV-NP using FuGENE HD transfection reagent according to the manufacture's instruction. Empty vector transfected cells were used as negative control antigen. At 48 hours post-transfection, cells were washed with PBS and lysed overnight in 4°C using cell lysis buffer consist of 50mM Tris pH 8.0, 300mM NaCl, 0.5% Triton X-100, supplemented with protease inhibitor cocktail (Thermo scientific). Lysate was cleared by centrifugation at 10000 x g for 10 mins at 4°C and stored in -80°C for further use.

## Sodium dodecyl sulfate-polyacrylamide gel electrophoresis (SDS-PAGE) and WB

Cell lysate containing NP was boiled at 95°C for 5 minutes with Laemmli buffer (Bio-Rad) and separated in 4–20% Mini-PROTEAN TGX Precast Protein Gels (Bio-Rad). Subsequently, the separated proteins were transferred to a polyvinylidene fluoride (PVDF) membranes (Merck Millipore) and blocked for 1 hour using PBS supplemented with 0.05% Tween 20 (PBST) containing 5% skimmed milk. The membrane was incubated with 1 µg/ml of mouse anti-LASV NP mAbs which were generated in house or commercial mAb from Zalgen (NP LASV MAb1474) was also used to detect LASV NP in the cell lysate. After washing three times with PBST, the membrane was incubated for 1 hour with horseradish peroxidase (HRP)-conjugated Goat anti-Mouse IgG (H + L) (1:10,000, Invitrogen) or Anti-rabbit IgG, HRP-linked Antibody (1:10,000, cell signaling technology), and protein bands were visualized using Immobilon Western Chemiluminescent HRP Substrate (Millipore) and images of blots were captured using Azure 300 (Azure biosystem).

## Generation of anti-LASV NP mAbs

The hybridomas secreting anti-LASV NP mAbs were generated as described previously [25]. Six weeks old female BALB/c mice were immunized intraperitoneally with 2 x 10^6 PFU of ML29. At one-month post-immunization, the mice were injected intraperitoneally with 100 µg of LASV Ojoko VLPs as a booster immunization. Three days later, the spleen was collected, splenocyte and P3U1 cells were fused and maintained according to standard procedure [25,26]. Approximately 2000 hybridoma candidates were screened using indirect ELISA with LASV Sauerwald NP as an antigen. Sixty-one positive candidates were further screened to assess their cross-reactivity with other LASVs from lineages I-IV. Single cell cloning of selected hybridoma candidates was performed by limiting dilution method. Cross-reactivity of each clone was

further confirmed by ELISA and ten clones were selected based on their cross-reactivity to each LASV lineages and used for further studies. The isotype class of each LASV NP mAbs were determined using mouse IgG isotyping kit (Biorad) following manufacturer's instructions. All animals were housed in animal biosafety level 2 (ABSL2) in GNL at UTMB.

### Indirect ELISA for mAb detection

ELISA plates (Thermo Fisher Scientific 442404) were coated with LASV NP lysates (1:100 dilution) in PBS overnight at 4°C. After blocking with 3% skimmed milk for 1 hour at room temperature (RT), hybridoma supernatants for hybridoma screening or 1 µg/mL of purified anti-LASV NP mAbs were incubated for 1 hour at RT. After three washes with PBST, HRP-conjugated Goat anti-Mouse IgG (H+L) (1:10,000, Invitrogen) was added into each well and incubated for 1 hour at RT. Plates were washed with PBST three times and immunoreactivity was visualized using 3, 3', 5, 5'-tetramethylbenzidine (TMB) substrate solution (Sigma). The reaction was stopped with 1 M phosphoric acid and the absorbance at 450 nm ($OD_{45}$) was recorded. $OD_{450}$ values were standardized with $OD_{450}$ of negative control. Positive binding of mAbs with LASV NPs was determined by comparing $OD_{450}$ values with negative control.

### Purification of mAbs

Single cell clones of each hybridomas were expanded in CD Hybridoma media (Gibco) to purify mAb for initial studies using Pierce Protein A Plus Agarose (Thermo Fisher Scientific) following manufacturer instructions. For the large-scale preparation, mAbs were purified from murine ascites fluid by Genescript, USA.

### HRP conjugation of mAbs and direct ELISA

The purified mAbs were conjugated with HRP using HRP conjugation kit (Abcam) as per manufacturer's instruction. HRP conjugation of mAbs was confirmed using direct ELISA, briefly, ELISA plates were coated with LASV Ojoko NP lysate overnight at 4°C, subsequently blocking and washing steps were performed as described in indirect ELISA section. Serially diluted HRP-conjugated mAbs were added to the wells and incubated for 1 hour. After washing, $OD_{450}$ values were measured as described in indirect ELISA section.

### Antibody-competitive ELISA

ELISA plates were coated with LASV Ojoko NP lysate (1:100 dilution) in PBS overnight at 4°C. After blocking with 3% skimmed milk for 1 hour at RT, 1 µg/mL of purified unconjugated anti-LASV NP mAbs were added at indicated concentrations and incubated for 1 hour at RT. After three washes with PBST, 1µg/ml HRP-conjugated anti-LASV NP mAbs were added into respective wells and incubated for 1 hour at RT. Washing and measurement of $OD_{450}$ is same as described in indirect ELISA section.

### Immunofluorescence assay (IFA)

HEK293T cells were transfected with each pC-LASV-NP using a FuGENE HD transfection reagent. Twenty-four hours post transfection, cells were washed with ice-cold PBST, subsequently fixed with 4% paraformaldehyde for 10 mins at RT and cells were permeabilized with PBS containing 0.5% Triton X-100 for 15 mins at 4°C. After washing with PBST, cells were blocked with PBS containing 3% bovine serum albumin for 1 hour at RT followed by incubation with the mouse 1µg/ml anti-LASV NP mAbs for 1 hour at RT. Cells were washed with PBST and then incubated with a 1:2000 diluted goat Alexa Fluor 488-conjugated anti-mouse IgG antibody (Thermo Fisher) and 1:5000 diluted Hoechst 33342 (Thermo Fisher) for 1 hour in the dark at RT. Images were acquired using a 20X objective lens in Olympus inverted microscope with Cell-Sense software (Olympus).

PLOS Neglected Tropical Diseases

## Antigen detection sandwich ELISA

ELISA plates were coated with 1 µg/mL LASV NP capturing mAbs overnight at 4°C. After blocking with 3% skimmed milk for 1 hour at RT, different antigens were added depending on assay: cell lysate containing LASV NP, live virus, serum samples from the guinea pigs infected with 100 PFU of LASV strain LF2384 [27] or mice infected with 100 PFU of LASV strain LF2350) [28], or VLPs for NP detection. For optimization of lysis buffer condition, 0.1% N-Lauroylsarcosine buffer (NLS buffer), 50mM Tris pH 8.0, 300mM NaCl and 1% Triton X-100 (TBS lysis buffer) and AVL buffer (Qiagen) were tested. For detection of live viruses, the viruses were serially diluted with DMEM and lysed with TBS lysis buffer by mixing 1:1 and incubated for 30 mins at RT for complete virus lysis. Lysed virus suspension was added as an antigen and incubated for 1 hour at RT. After five washes with PBST, 1 µg/mL of HRP-conjugated mouse anti-LASV NP mAbs were added into each well and incubated for 1 hour at RT. Washing and measurement of $OD_{450}$ is same as described in indirect ELISA. The results were determined as positive or negative using a threshold value, mean plus three standard deviations of the negative control.

## Statistical analysis

Statistical evaluations were performed using a one-way analysis of variance (ANOVA), supplemented with Turkey's multiple-comparison test for individual comparisons. For grouped data analyses, a two-way ANOVA was applied, followed by Dunnett's multiple-comparison test for post-hoc analysis. All statistical analyses were conducted using GraphPad Prism software (ver. 10).

## Results

### Characterization and evaluation of cross-reactive anti-LASV NP mAbs

Following initial screening by indirect ELISA, a total of 10 purified mAbs were obtained, all demonstrating cross-reactivity with LASVNP from lineages I to VII (Fig 1A). Among these, mAbs 28-2-3, 27–2, 30–12, 40–3, 57–2, and 35–2 exhibited broad cross-reactivity to all LASV lineages in ELISA. In contrast, mAbs 3-2-3, 36-2-1, and 51-1-2 lacked reactivity to LASV NP from lineages IV, VI, and I, respectively. Additionally, mAb 12-1-1 showed non-specific binding to negative control cell lysates and was therefore excluded from further analysis. The binding profiles of the remaining mAbs to LASV NPs across different lineages were further evaluated using WB. Most of the mAbs detected a 66-kDa protein band consistent with the molecular mass of LASV NP (Figs 1B and S1) [29]. Notably, these novel mAbs also demonstrated more reliable and sensitive detection of all the lineages compared to the commercially available Ab, which failed to detect lineage II and partially detected lineage IV in WB (S1 Fig). We also characterized these mAbs using IFA (Figs 1C and S3). The reactivity in ELISA, WB, IFA and isotype of mAbs are summarized in Table 1. Based on these results, mAbs 27–2, 28-2-3, 30–12, 40–3, and 57–2 were used for further evaluations.

### Competitivity of each mAb against LASV NP

To evaluate epitope competition among the mAbs targeting LASV NP, we performed an antibody-competition ELISA. Successful HRP-conjugation of each mAb was first confirmed using direct ELISA (S2 Fig), and the optimal concentration of each HRP-conjugated mAb was determined to achieve an OD value between 0.5 and 1. Antibody-competition ELISA results demonstrated that mAb 28-2-3 recognizes a distinct, non-competing epitope that does not overlap with the binding sites of the other mAbs tested. (Fig 2). Since mAb 28-2-3 showed non-competitivity to other mAbs, we exploited this feature to develop antigen-capture sandwich ELISA for LASV detection. We also selected some competitive pairs of mAbs to further evaluate their compatibility in sandwich ELISA.

### Development of antigen-capture sandwich ELISA using anti-LASV NP mAbs

To develop an antigen-capture sandwich ELISA, we first optimized the viral lysis conditions by evaluating several disruption buffers (Fig 3A). Buffer selection was based on three criteria: effective viral lysis, availability in LASV-endemic regions,

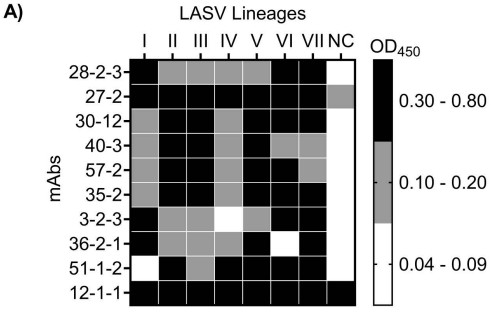
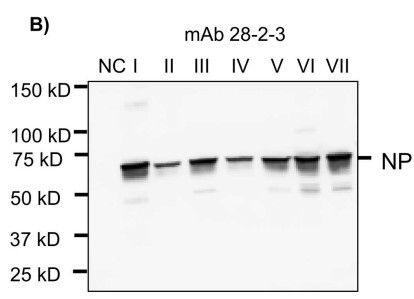
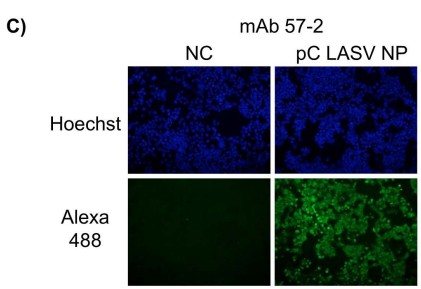

**Fig 1. Generation of mAbs for LASV NP. A)** Cross-reactivity of mAbs to different LASV lineages NP was measured using cell lysate in indirect ELISA. $OD_{450}$ for each mAb is summarized as heat map. **B)** Representative of binding of anti-LASV NP mAb with cell lysate of HEK293T expressing LASV NPs (lineage I-VII) using WB, each lysate was diluted 1:100 times and used as an antigen for measurement of binding of mAb 28-2-3. **C)** Representative images of IFA showing binding of mAb 57-2 to the HEK293T cells expressing LASV NP, 24 hours post transfection cells were fixed, permeabilized and incubated with mAb 57-2 and detected by Alexa488 anti-mouse IgG antibody. Hoechst was used for nucleus staining. HEK293T cells transfected with pCAGGS empty vector were used as negative control (NC).

**Table 1. Binding profile of mAbs against LASVs belonging to Lineage I-VII.**

| mAb | Isotype | ELISA | | | | | | | WB | | | | | | | IFA | | | | | | |
|---|---|---|---|---|---|---|---|---|---|---|---|---|---|---|---|---|---|---|---|---|---|---|
| | | I | II | III | IV | V | VI | VII | I | II | III | IV | V | VI | VII | I | II | III | IV | V | VI | VII |
| 3-2-3 | IgG1 (κ) | + | + | + | + | ± | + | + | − | + | + | + | + | − | − | NT | | | | | | |
| 12-1-1 | IgG1 (κ) | + | + | + | + | + | + | + | + | + | + | + | + | + | + | NT | | | | | | |
| 27-2 | IgG2a (κ) | + | + | + | + | + | + | + | + | + | + | + | + | + | + | + | + | + | + | + | + | + |
| 28-2-3 | IgG2b (λ) | + | + | + | + | + | + | + | + | + | + | + | + | + | + | − | − | − | − | − | − | − |
| 30-12 | IgG1 (κ) | + | + | + | + | + | + | + | + | + | + | + | + | + | + | + | + | + | + | + | + | + |
| 35-2 | IgG1 (κ) | + | + | + | + | + | + | + | + | + | + | + | + | + | + | + | + | + | + | + | + | + |
| 36-2-1 | IgA (κ) | + | + | + | + | + | ± | + | − | − | − | − | − | − | − | NT | | | | | | |
| 40-3 | IgG2a (κ) | + | + | + | + | + | + | + | + | + | + | + | + | + | + | + | + | + | + | + | + | + |
| 51-1-2 | IgG1 (κ) | − | + | + | + | + | + | + | − | − | − | − | − | − | − | NT | | | | | | |
| 57-2 | IgG1 (κ) | + | + | + | + | + | + | + | + | + | + | + | + | + | + | + | + | + | + | + | + | + |

'+' indicates positive binding, '-' indicates no binding, '+/-' indicates ambiguous data and 'NT' indicates not tested in IFA studies.

and compatibility with ELISA applications. For optimization, we used ML29, a Lassa fever (LF) vaccine candidate that contains the full S-segment (encoding NP and GPC) of the LASV Josiah strain and can be safely handled under BSL-2 conditions. Our results indicated that TBS lysis buffer consistently produced significantly higher OD450 values compared to other tested buffers. Based on these findings, TBS lysis buffer was selected for use in subsequent sandwich ELISA

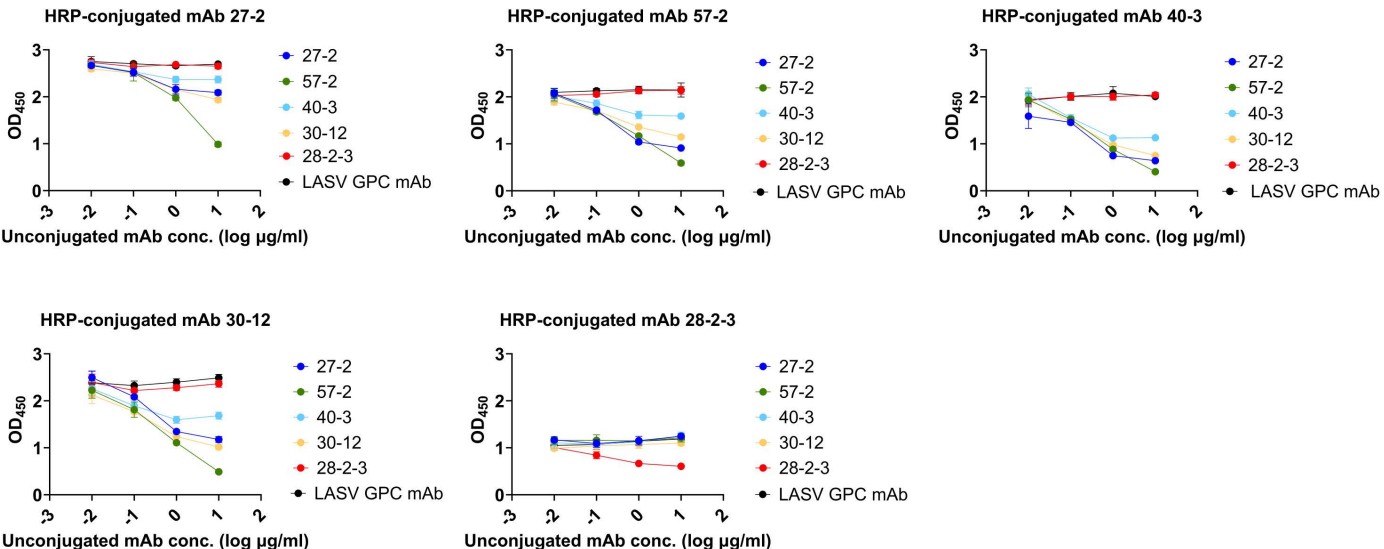

**Fig 2. Ab competitive ELISA.** The cross-competition profile of mAbs for binding to LASV NP was determined using Ab competitive ELISA. HEK293T cell lysate expressing LASV lineage III (Ojoko) NP was diluted to 1:100 in PBS and coated on the ELISA plate surface. Serially diluted unconjugated mAbs were used as primary antibody as represented in different colors in each graph and then 1μg/ml of HRP-conjugated mAbs for LASV NP were used as secondary antibody for each measurement. Anti-LASV GPC mAb was used as an IgG control.

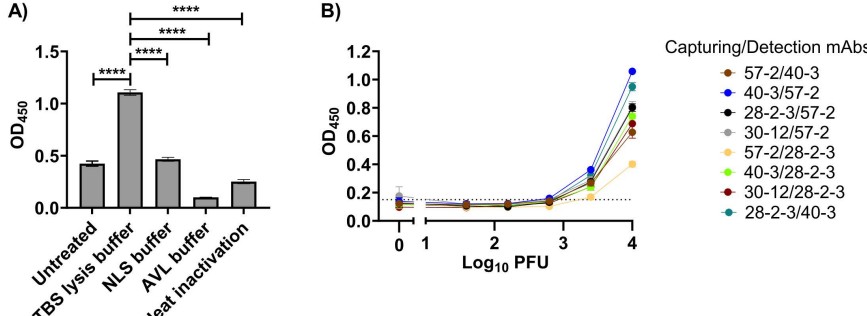

**Fig 3. Development of sandwich ELISA using LASV NP mAbs. A)** Comparative analysis of virus lysis condition for efficient NP detection. $1 \times 10^4$ PFU/ml ML29 was treated with either different virus lysis buffers or heat inactivated, then $OD_{450}$ values measured by sandwich ELISA using 40-3 as capturing and 57-2 as detection mAbs. Data were analyzed using one-way ANOVA, followed by Turkey's test for multiple comparisons, where **** $P < 0.0001$. **B)** Determination of mAb pairs of sandwich ELISA for detection of live virus. Serially diluted ML29 were used to measure the sensitivity of sandwich ELISA with different mAb pairs.

experiments. (Fig 3A). Furthermore, using ML29, we estimated detectable virus titer of authentic LASVs in our sandwich ELISA. The results showed that the limit of detection (LOD) for ML29 ranged from $6.2 \times 10^2$ to $2.5 \times 10^3$ PFU, depending on the mAb pair used (Table 2 and Fig 3B). The combination of mAb 57–2 as the capture antibody and mAb 28-2-3 as the detection antibody required a minimum viral titer of approximately $1.0 \times 10^4$ PFU for reliable detection and was therefore excluded from further consideration for authentic LASV detection. Based on the LODs and $OD_{450}$ values, we selected four optimal mAb pairs, highlighted in bold in Table 2. These included combinations using either 40–3 or 30–12 as the capture antibody paired with 28-2-3 as the detection antibody, as well as combinations using 40–3 or 28-2-3 as the capture antibody with 57–2 as the detection antibody.

**Table 2. LOD of each mAb combination for ML29.**

| mAb combination | | LOD (PFU) |
|---|---|---|
| **Capturing** | **Detection** | |
| 57-2 | 40-3 | 2.5 x10³ |
| 28-2-3 | 40-3 | 2.5 x10³ |
| 57-2 | 28-2-3 | 1.0 x10⁴ |
| **40-3** | **28-2-3** | 2.5 x10³ |
| **30-12** | **28-2-3** | 6.2 x10² |
| 30-12 | 57-2 | 2.5 x10³ |
| **40-3** | **57-2** | 2.5 x10³ |
| **28-2-3** | **57-2** | 2.5 x10³ |

### Detection of LASV belonging to Lineage I-VII using sandwich ELISA

The detection efficiency of the developed sandwich ELISA was evaluated using authentic LASV strains in a BSL-4 laboratory. LASV strains representing multiple lineages-including Pinneo (lineage I), Sauerwald (lineage II), Ojoko (lineage III), and LF2384 (lineage IV)-were tested using various mAb combinations. Each LASV strain was serially diluted to determine the limit of detection (LOD) of the assay for each lineage (Fig 4A and Table 3). The mAb pairs 30–12/28-2-3, 40–3/28-2-3, and 28-2-3/57–2 consistently demonstrated limits of detection (LOD) ranging from $6.2 \times 10^2$ to $2.5 \times 10^3$ PFU across the tested LASV strains. These LOD values were comparable to those observed in our ML29-based assays. The mAb pair 40–3/57–2 showed slightly higher LODs for LASV lineages I, II, and III, though still within the same logarithmic range. These results indicate that the developed sandwich ELISA utilizing novel anti-LASV NP mAbs is effective for LASV detection. Due to the unavailability of authentic LASV strains from lineages V-VII, VLPs corresponding to these lineages were generated for further evaluation. As a positive control, VLPs from LASV lineage IV were also included in our experimental setup. Sandwich ELISA results suggested these mAbs showed cross-reactivity against LASV VLPs from lineage IV-VII (Fig 4B). We determined the LOD of this assay for LASV VLPs to be approximately 150 ng. Based on the similar LODs and binding characteristics of the mAbs pairs to lineage IV VLPs, we extrapolated the LODs for lineages V, VI, and VII. The estimated virus LOD for these lineages was approximately $6.2 \times 10^2$ PFU, suggesting a comparable level of sensitivity across these divergent LASV lineages. Furthermore, to evaluate the analytical performance of the sandwich ELISA with biological matrix, we tested it using serum samples from LASV-infected guinea pigs and mice. Among the tested mAb pairs, the combination of 28-2-3 as the capture mAb and 57–2 as the detection mAb showed the most robust and consistent detection of LASV in guinea pig and mice serum samples (Table 4). Although antigen dynamics in rodent models may not perfectly mirror human infection, these data validate the assay's ability to detect LASV NP in complex sera from active infection animal models, representing a critical step toward establishing future clinical diagnostic development.

Additionally, to rule out the false positive detection, rVSV was used as negative control. We also measured binding of different mAbs pair with EBOV, MARV, and LCMV as a representative for causative agents of viral hemorrhagic fevers (Fig 4C). As expected, all pairs of mAbs (capturing and detection mAbs) 30–12/28-2-3, 40–3/28-2-3, 28-2-3/57–2, and 40–3/57–2, did not bind these viruses.

### Discussion

We successfully generated mAbs targeting LASV NP with cross-lineage reactivity. The binding specificity and performance of these novel mAbs were further validated using multiple immunoassays, including WB and immunofluorescence assay (IFA). The NP of arenaviruses is an ideal target for diagnostic assays due to its high abundance within virions, elevated antigenemia in clinical infections, and strong immunogenicity [30].

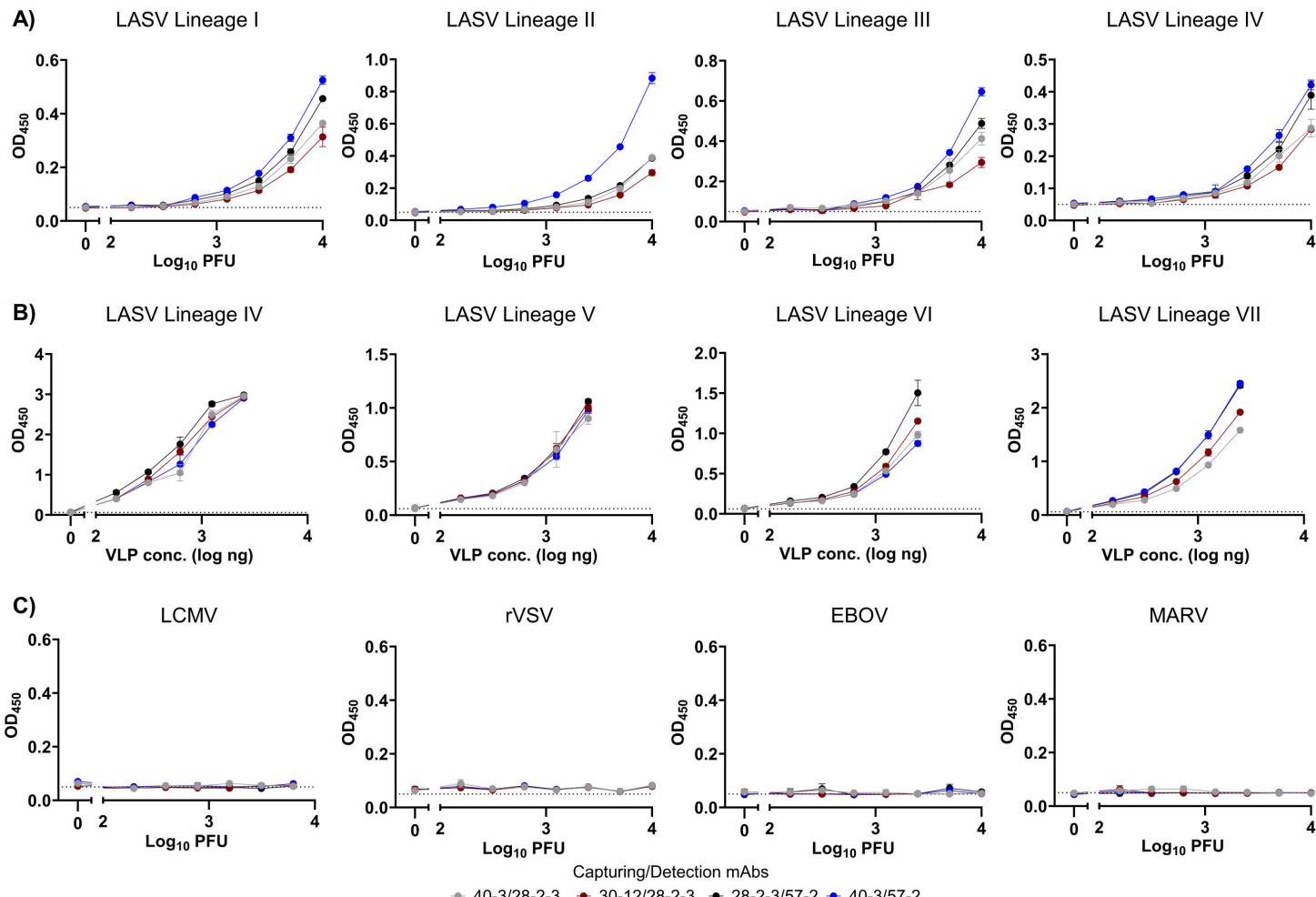

**Fig 4. Detection of *pan*-LASV using dual mAbs-based sandwich ELISA. A)** $2x10^5$ PFU/ml of LASV lineages I-IV were serially diluted and used as an antigen for sandwich ELISA with 1μg/ml of each capturing and detection mAbs. Binding of different pairs of capturing mAb/detection mAb with LASV NP are denoted in the graph as $OD_{450}$. **B)** Sandwich ELISA for detection of LASV NP from lineages V-VII were measured using VLPs, lineage IV VLPs was used as positive control for this measurement. **C)** Measurement of NP mAbs binding with rVSV, EBOV, MARV, and LCMV at different concentration of authentic virus using different combinations of mAbs. Data represents means and standard deviations (SDs) of $OD_{450}$ from three biological triplicates. The threshold of sandwich ELISA was calculated as the mean+3SDs of $OD_{450}$ value of negative control and represented as a black dotted horizontal line in the graphs.

**Table 3. LOD of each mAb combination for LASVs (Lineage I-IV).**

| mAb combination | | LOD (PFU) | | | |
|---|---|---|---|---|---|
| Capturing | Detection | Lineage I | Lineage II | Lineage III | Lineage IV |
| 40-3 | 28-2-3 | $6.2x10^2$ | $1.2x10^3$ | $1.5x10^2$ | $6.2x10^2$ |
| 30-12 | 28-2-3 | $6.2x10^2$ | $3.1x10^3$ | $6.2x10^2$ | $6.2x10^2$ |
| 40-3 | 57-2 | $1.2x10^3$ | $3.1x10^3$ | $1.2x10^3$ | $6.2x10^2$ |
| 28-2-3 | 57-2 | $6.2x10^2$ | $3.1x10^3$ | $2.5x10^3$ | $6.2x10^2$ |

**Table 4. LASV detection from serum in LASV-infected guinea pigs and mice.**

| mAb combination | | Guinea pig | | | | Mouse | | | |
|---|---|---|---|---|---|---|---|---|---|
| Capturing | Detection | GP#1 | GP#2 | GP#3 | GP#4 | M#1 | M#2 | M#3 | M#4 |
| 40-3 | 28-2-3 | Neg | Neg | Neg | Neg | Neg | Neg | Neg | Neg |
| 30-12 | 28-2-3 | Neg | Neg | Neg | Neg | Neg | Neg | Neg | Neg |
| 40-3 | 57-2 | Neg | Neg | **Pos** | Neg | Neg | **Pos** | Neg | Neg |
| 28-2-3 | 57-2 | **Pos** | Neg | **Pos** | Neg | Neg | **Pos** | **Pos** | Neg |
| Virus titer (PFU/mL) | | 800 | 380 | 31000 | NC | 1100 | 15000 | 2900 | NC |

Neg: negative, Pos: positive, NC: negative control (uninfected animals)

The samples were collected at 11 (GP#1 and GP#2), 17 (GP#3), or 60 (M#1–3) days post-infection.

Rapid and accurate diagnosis of LASV infection is critical not only for endemic regions but also for global public health, as the rising number of imported cases poses a growing international concern [31]. Several commercial and laboratory-based diagnostics, such as RT-PCR-based detection methods have been developed, but these tests fail to detect all reported lineages of LASV in a single assay due to the genetic diversity of LASV [32]. In addition, several RT-PCR assays such as GPC RT-PCR/1994 and GPC RT-PCR/2007 have also been reported to yield false negative results [33]. In fact, assays that claim *pan*-LASV detection have been shown to reliably detect only lineages II, III, and IV, and are therefore not recommended for diagnostic use or as screening tools for suspected LASV cases [16–18,34]. In contrast, our antigen-capture sandwich ELISA offers a robust alternative that addresses the inter-lineage heterogeneity of LASV. Although the sensitivity of antigen detection ELISA is lower than RT-PCR-based detection, we demonstrated robust cross-reactivity against LASV belonging to all known lineages. Infrastructure, such as stable electricity, is a concern in the LASV-endemic area of, as it affects the effectiveness and timeliness of diagnosis. Our mAbs can be further utilized to develop more cost-effective, simple, and rapid diagnostic systems such as lateral flow immunochromatography.

The non-competitive binding of mAb 28-2-3 relative to other mAbs indicates that it targets a distinct epitope on the LASV NP. Interestingly, we found that sandwich ELISAs using combinations of competitive antibodies demonstrated equal or even lower LODs compared to those using non-competitive mAb pairs. This suggests that, despite targeting overlapping or adjacent epitopes, these competitive mAbs can still bind simultaneously to the NP. These results align with the known oligomeric structure of LASV NP, which likely presents multiple accessible binding sites [35]. This allows multiple antibody molecules to bind to different monomeric NP within the oligomeric complex.

The LODs of our antigen-capture sandwich ELISA were around $10^3$ PFU for LASV lineages I-IV. This is equivalent to approximately $10^4$ PFU/ml in a serum sample, which is sufficient to detect viral load in the clinically apparent cases of LF. Given that viremia in LASV infections among febrile patients has ranged up to $10^8$ $TCID_{50}$/ml [36], these results suggested that this sandwich ELISA can be utilized for the early diagnosis of the LASV infections in clinical samples. The robustness of our assay requires further validation using clinical samples, as factors such as sample viscosity and the presence of potential assay inhibitors may affect overall performance. Our findings show that the mAbs pair consisting of 28-2-3 and 57–2 facilitate sensitive detection of LASV, confirming that sandwich ELISA can detect the virus in serum samples even at very low viral titers.

However, systematic evaluation across diverse sample types and clinical conditions will therefore be essential to establish reliability under real-world diagnostic settings. Moreover, comparative analysis with widely used diagnostic methods for LF, such as RT-PCR, is needed to establish sandwich ELISA assay performance. Direct comparison of analytical sensitivity, specificity, LOD and overall diagnostic accuracy will be critical to determine the relative utility of the sandwich ELISA employing the newly developed mAbs. Such studies will help define its potential role as a complementary or alternative tool in routine LF diagnostics.

While this study successfully demonstrated broad cross-reactivity across LASV lineages, it is also important to assess potential cross-reactivity with other arenaviruses, given the high sequence homology between LASV NP and New World arenavirus NPs, which share approximately 21% identity and ~80% similarity [37]. The mAbs generated against LASV NP may potentially cross-react with NPs of other arenaviruses, which may lead to false-positive reactions in non-LASV cases [5]. From a structural perspective, the homology suggests a potential for cross-reactivity; however, the distinct geographical distribution of these viruses significantly limits the risk of false-positive diagnostic results in clinical practice. Although our antigen detection Sandwich ELISA did not detect LCMV, the detection methods using these novel mAbs can potentially detect LASV-like arenaviruses that cause viral hemorrhagic fever [38].

Overall, in this study, we have generated cross-reactive anti-LASV NP mAbs and developed experimental antigen-capture sandwich ELISAs, which offer strong potential for *pan*-LASV detection methods that can be utilized in clinical settings.

## Supporting information

**S1 Fig. Reactivity of mAbs against LASV NPs by western blotting.**
(DOCX)

**S2 Fig. HRP conjugation of LASV NP mAbs.**
(DOCX)

**S3 Fig. Reactivity of mAbs against LASV NPs by immunofluorescent assay.**
(DOCX)

## Author contributions

**Conceptualization:** Junki Maruyama.

**Funding acquisition:** Slobodan Paessler, Junki Maruyama.

**Investigation:** Ruchi Paroha, Takeshi Saito, Shintaro Yamada, Christine Click.

**Methodology:** Takeshi Saito, Junki Maruyama.

**Supervision:** Junki Maruyama.

**Writing – original draft:** Ruchi Paroha.

**Writing – review & editing:** Takeshi Saito, Shintaro Yamada, Christine Click, Slobodan Paessler, Junki Maruyama.

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
