## [Decision Letter · Decision Letter 0]

3 Sep 2025

PNTD-D-25-01122

Development of novel monoclonal antibodies for detection of pan-Lassa virus

Dear Dr. Maruyama,

Thank you for submitting your manuscript to PLOS Neglected Tropical Diseases. After careful consideration, we feel that it has merit but does not fully meet PLOS Neglected Tropical Diseases's publication criteria as it currently stands. Therefore, we invite you to submit a revised version of the manuscript that addresses the points raised during the review process.

Please submit your revised manuscript within 60 days Nov 02 2025 11:59PM. If you will need more time than this to complete your revisions, please reply to this message or contact the journal office at plosntds@plos.org. Please include the following items when submitting your revised manuscript:

We look forward to receiving your revised manuscript.

Kind regards,

Katharina Röltgen

Academic Editor

Michael Holbrook

Section Editor

Shaden Kamhawi

co-Editor-in-Chief

Paul Brindley

co-Editor-in-Chief

**Additional Editor Comments:**

We have received comprehensive review comments on your manuscript from three experts in the field. All three reviewers agreed that the work on the development of assays to broadly detect different Lassa virus lineages is of general interest and of clear public health relevance. At the same time, they raised similar major concerns about the manuscript that will need to be addressed before it can be considered further. The most significant issues relate to the incomplete validation of the assays and presentation of data, as detailed below:

1. Validation in a clinical context: the reviewers emphasized the lack of validation of the developed mAb-based assays using patient samples from endemic areas, as well as the absence of appropriate positive and negative controls. Without these experiments, it is difficult to realistically assess assay performance. As noted, strong results in laboratory-based assays do not necessarily translate to adequate accuracy in clinical settings and more evidence of the clinical relevance of the assay should be provided. Assays should be validated with clinical samples.

2. Overstated conclusions: In its current form, the manuscript presents conclusions that appear stronger than the data support. These claims should be carefully tempered to remain within the scope of the evidence provided and limitations of the data should be acknowledged in more detail.

3. Incomplete presentation of mAb data: the reviewers noted that data presented in several Figures are only based on a subset of mAbs, which they found insufficient. A comprehensive presentation of data for all relevant mAbs is required.

**Journal Requirements:**

1) We do not publish any copyright or trademark symbols that usually accompany proprietary names, eg ©,  ®, or TM  (e.g. next to drug or reagent names). Therefore please remove all instances of trademark/copyright symbols throughout the text, including:

- ® on page: 7.

3) Tables should not be uploaded as individual files. Please remove these files and include the Tables in your manuscript file as editable, cell-based objects. For more information about how to format tables, see our guidelines:

https://journals.plos.org/plosntds/s/tables

**Reviewers' Comments:**

Reviewer's Responses to Questions

**Key Review Criteria Required for Acceptance?**

**Methods:**

-Are the objectives of the study clearly articulated with a clear testable hypothesis stated?

-Is the study design appropriate to address the stated objectives?

-Is the population clearly described and appropriate for the hypothesis being tested?

-Is the sample size sufficient to ensure adequate power to address the hypothesis being tested?

-Were correct statistical analysis used to support conclusions?

-Are there concerns about ethical or regulatory requirements being met?

Reviewer #1: Objectives are clearly articulated

Study design is appropriate but incomplete as noted below.

Correct statistical analyses used for data presented.

No ethical concerns.

Reviewer #2: Paroha et al. present an interesting study describing the generation and characterization of new monoclonal antibodies against LASV NP from multiple lineages, followed by the development of an antigen-capture sandwich ELISA intended for Lassa fever diagnosis.

The methods for antibody generation, screening, and binding characterization (ELISA, Western blotting, and immunofluorescence), as well as identification of non-competing mAb pairs using competitive ELISA, are logical, thorough, and clearly described. The subsequent design of the sandwich ELISA and testing against recombinant antigens, ML29 spiked in serum, LASV isolates from lineages I–IV, and LASV VLPs from lineages V–VII are appropriate for the authors’ stated objectives.

That said, the absence of validation using samples from LASV-infected hosts is a major limitation of the methodological framework. While access to acute Lassa fever patient samples is understandably challenging, the authors do appear to have BSL-4 laboratory capacity, and validation using sera from infected animals (e.g., from ongoing or archived challenge studies) would not seem infeasible. Such experiments would provide a more realistic assessment of the assay’s performance in the presence of an active host immune response, clarify whether circulating antibodies might interfere with NP capture, and yield a more accurate estimate of the detection limit. In addition, comparison of assay performance to RT-qPCR would address key questions of sensitivity and specificity. Without these elements, the potential clinical utility of the approach remains uncertain.

Reviewer #3: Study objectives are clearly stated. The study design does not fully address the stated objective. Statistical details are lacking. See overall comments for details.

**Results:**

-Does the analysis presented match the analysis plan?

-Are the results clearly and completely presented?

-Are the figures (Tables, Images) of sufficient quality for clarity?

Reviewer #1: Results, including figures, are appropriate except as noted below.

Reviewer #2: The analyses presented are consistent with the described methods and are explained clearly and concisely. The results are well structured, and the figures and tables are of good quality, with clear labeling and appropriately detailed legends that make the data easy to interpret. Overall, the presentation of the results is strong.

I have two minor suggestions for improvement:

In Figure 3C (lines 304–307), if multiple antibody pairs were tested against ML29-spiked serum, it may be clearer to display the results from all pairs, as in Figure 3B, rather than only showing the 40-3 + 57-2 pair. This would allow the reader to better evaluate relative assay performance.

In the Discussion (lines 401–405), the authors mention western blot results assessing cross-reactivity of the mAbs against other arenaviruses, but these data are not shown. While this experiment is peripheral to the main objectives, including the blots in the Supplementary Information (if permissible) would provide completeness and transparency.

Reviewer #3: The analysis matches the objectives, but is incomplete. Comprehensive presentation of data is lacking. The figures and tables are of sufficient quality.

**Conclusions:**

-Are the conclusions supported by the data presented?

-Are the limitations of analysis clearly described?

-Do the authors discuss how these data can be helpful to advance our understanding of the topic under study?

-Is public health relevance addressed?

Reviewer #1: Conclusions are appropriate. Limitations are well noted. Public health relevance is addressed.

Reviewer #2: The development of improved diagnostics for Lassa fever remains an important and unmet public health need, particularly for tools that can be deployed in resource-limited community clinics and at the point of care. This study contributes meaningfully by identifying novel broadly reactive monoclonal antibodies and demonstrating their application in an antigen-capture ELISA format. The findings represent a potentially valuable step toward diagnostic tools with utility across diverse LASV lineages.

At the same time, the conclusions somewhat overstate the likely impact of the assay as presented. Other groups have previously developed antigen-capture ELISAs and rapid diagnostic tests (RDTs) for LASV that performed well in controlled laboratory studies but ultimately showed inadequate sensitivity and specificity in field evaluations—limitations that explain why RT-qPCR remains the diagnostic gold standard in endemic regions. The authors briefly cite these mixed outcomes, but they do not provide a clear rationale for why their assay is likely to perform differently or better in real-world settings. For example, most clinical evaluations of Zalgen’s ReLASV Pan-Lassa RDT were conducted in Nigeria and Sierra Leone, where lineages II, III, and IV predominate. Thus, the poor field performance of that test was unlikely due to restricted lineage coverage. More broadly, antigen-based assays face inherent constraints, such as reduced sensitivity later in infection when circulating antibodies begin to complex with free antigen. This context underscores that broader cross-lineage detection, while valuable, may not itself overcome the central challenges of sensitivity and specificity in clinical practice (see, for example, Chi et al., PLoS Glob Public Health, 2025; Elsinga et al., Lancet Infect Dis, 2024; Boisen et al., Sci Rep, 2020; Boisen et al., Sci Rep, 2018).

To strengthen their conclusions, I suggest that the authors explicitly acknowledge these limitations and outline the steps needed to establish the assay’s utility. In particular, side-by-side comparisons of their mAbs against existing antigen-capture platforms, coupled with validation against RT-qPCR (the diagnostic gold standard), will be essential. Such experiments, ideally performed using clinical or animal challenge samples, would provide critical evidence that this assay represents a meaningful improvement over current tools.

Reviewer #3: The conclusions are not fully supported by the data as presented. The limitations are only moderately described.

**Editorial and Data Presentation Modifications?**

Reviewer #1: Minimal line editing needed.

Reviewer #2: The manuscript is generally well organized, and the figures and tables are clear and appropriately annotated. However, there are several places across the manuscript where grammar, syntax, and sentence structure issues make the text difficult to follow. I suggest that the authors carefully review the full manuscript to address these issues and improve clarity and readability, which would strengthen the overall presentation of their findings.

Reviewer #3: (No Response)

**Summary and General Comments:**

Reviewer #1: In their manuscript entitled, “Development of novel monoclonal antibodies for detection of pan-Lassa virus”, Paroha and colleagues describe the generation of anti-LASV NP mAbs with cross reactivity for all seven LASV lineages and the development of an antigen-capture sandwich ELISA for pan-LASV detection. An unknown number of mice were immunized with virus like particles. Approximately 2000 hybridomas were created and screened. Of these, 61 screened candidates were tested for reactogenicity to LASV NP from all seven lineages. Of these, 10 proved cross-reactive. Competition assays were performed to identify mAbs with non-overlapping epitopes. An antigen-capture sandwich ELISA was then developed using optimized pairs of non-completing mAbs.

Overall, this is a very well written manuscript. The methods and results are very clearly written and understandable. As noted below, I have some minor comments that should be addressed. My greatest concern is the lack of positive and negative controls mentioned in the text. While the ELISA development was clearly described, it is not useful without positive controls. Furthermore, the authors have yet to describe the next critical step in the assays development which is testing it using human samples from endemic areas from LASV suspected patients with both positive and negative results by another methodology (e.g. pcr). Even data using a panel of retrospective samples would be adequate to give an initial determination of sensitivity and specificity. As written, the mAb and assay development are incomplete.

Other Comments:

1. Methods: Overall, the description of the methods is quite complete and well written. However, in the description of the ELISAs and immunofluorescence assays there is no mention of positive or negative controls. What was used? Please describe these as well as the number of replicates used in each experiment and the reproducibility of the results.

2. Why were 4 of the mAbs not tested by IFA?

3. Was a non-LASV mAb used as a negative control to ensure no non-specific binding was occurring?

4. Line 306: What serum was used? Human or mouse?

5. Lines 341-343: I understand why rVSV, EBOV and MARV were used to identify non-specific binding. It would also be useful to know if there is binding to LCMV.

6. Line 383: The word sensitivity should not be used here as no studies to determine sensitivity were presented. LOD would be more appropriate.

7. Lines 396 -408: The potential cross-reactivity with other arenaviruses should be moved to the results section and the data presented. LCMV should be included as well.

Reviewer #2: This study reports the generation of broadly reactive monoclonal antibodies against LASV nucleoprotein and their application in a sandwich ELISA. The work is carefully executed, the experiments are appropriate to the stated objectives, and the data are presented clearly. These reagents could make a useful contribution to ongoing efforts to develop improved diagnostics for Lassa fever, which remains a pressing public health need.

The principal limitation is the absence of validation with serum or blood samples from LASV-positive hosts. While the binding results with purified virus isolates and VLPs are convincing, there is ample published evidence showing that strong performance in controlled laboratory assays does not necessarily translate into adequate sensitivity and specificity in clinical or field settings. Without testing in animal or patient samples—or a much clearer rationale for why this assay is likely to perform better than previous tools—the broader significance of the findings is difficult to assess.

In its current form, the study offers valuable preliminary data and novel reagents. However, to substantiate the conclusions, additional testing (discussed above) is strongly encouraged. If such experiments are not feasible at this stage, the conclusions should at minimum be framed more cautiously, with greater emphasis on the next steps needed to establish clinical relevance and real-world utility.

Reviewer #3: Paroha, et al, address the need for improved diagnostic methods for Lassa Fever (LF), which are currently complicated by the genetic diversity of the virus. The authors generated a panel of monoclonal antibodies (mAbs) targeting the LASV nucleoprotein (NP). Several mAbs showed broad reactivity across LASV lineages I-VII in ELISA and western blot and immunofluorescence assays. Antigen-capture assays were then used to identified pairs of mAbs that could be used in diagnostics. While the premise is reasonable, the lack of comprehensive data presentation and controls, and unjustified conclusions about the sensitivity and clinical applicability is a major weakness. Without addressing these concerns, the impact of the study and the potential of the developed ELISA for improved LF diagnosis remain uncertain.

Key concerns include:

• There is incomplete data presentation in Figs 1, S1, S2 and Fig 3. Data is shown for only a subset of the mAbs. The authors should include comprehensive data presentation for all mAbs, or at the minimum, those that are bolded in Table 1. Only one mAb pair is presented in Fig 3. It is unclear whether the other mAbs pairs were also tested in this format or if they did not perform well. Regardless, all data should be presented.

• Critical details are missing regarding ELISA data analysis, including the quantification method for lysates to ensure equal coating of plates, the basis for LOD determination and controls used to set this LOD. Controls in general are lacking throughout the experiments.

• Reactivity levels are misrepresented in Table 1. For example, mAb 57-2's western blot reactivity to LI, LVI, and LVII is weaker than to LII-V, but all are indicated as "+," giving a false impression of uniform reactivity.

• The signal for the mAb pair 30-12/282-3 is at the LOD line. The data presented don't support determination of 6.2E2 as the sensitivity for this pair.

• The lineage-specific data presented in Figure 4 is focused on virus diluted in serum-free DMEM when viral-spiked sera controls would have been more impactful for measuring clinical relevance. The signal for the ML29-reassortment is reduced from a maximum of 1.2 in the absence of sera to 0.5 in the presence of sera. The maximum signals for live LASV were 0.4-0.9 in the absence of sera. Presumably, the same reduction in signal would occur in the presence of sera, calling into question the true sensitivity for live Lassa.

• Line 305-307: The authors claim of “reliable detection of virus in sera” based on a single replicate with one mAb pair is an overstatement based on the limited analyses and replication of experiments.

• The authors state that “… the detection methods using these novel mAbs can detect unknown LASV belonging to another lineage or LASV-like arenaviruses that cause viral hemorrhagic fever”. The data supporting this claim is not presented. If the authors mean that sequence conservation suggests the mAbs might detect LASV-like arenaviruses, then the statement needs clarification.

PLOS authors have the option to publish the peer review history of their article (what does this mean?). If published, this will include your full peer review and any attached files.

Reviewer #1: No

Reviewer #2: No

Reviewer #3: No

**Figure resubmission:**
---

## [Decision Letter · Decision Letter 1]

21 Apr 2026

PNTD-D-25-01122R1Development of novel monoclonal antibodies for detection of pan-Lassa virusPLOS Neglected Tropical Diseases Dear Dr. Maruyama, Thank you for submitting your manuscript to PLOS Neglected Tropical Diseases. After careful consideration, we feel that it has merit but does not fully meet PLOS Neglected Tropical Diseases's publication criteria as it currently stands. Therefore, we invite you to submit a revised version of the manuscript that addresses the points raised during the review process. Please submit your revised manuscript by May 21 2026 11:59PM. If you will need more time than this to complete your revisions, please reply to this message or contact the journal office at plosntds@plos.org.  Please include the following items when submitting your revised manuscript: * A letter that responds to each point raised by the editor and reviewer(s). You should upload this letter as a separate file labeled 'Response to Reviewers'. This file does not need to include responses to any formatting updates and technical items listed in the 'Journal Requirements' section below.* A marked-up copy of your manuscript that highlights changes made to the original version. You should upload this as a separate file labeled 'Revised Manuscript with Track Changes'.* An unmarked version of your revised paper without tracked changes. You should upload this as a separate file labeled 'Manuscript'.

We look forward to receiving your revised manuscript.

Kind regards,

Katharina Röltgen

Academic Editor

PLOS Neglected Tropical DiseasesMichael HolbrookSection EditorPLOS Neglected Tropical Diseases

Shaden Kamhawi

co-Editor-in-Chief

Paul Brindley

co-Editor-in-Chief

**Reviewers' comments:**

 Reviewer's Responses to Questions

**Key Review Criteria Required for Acceptance?**

**Methods**

-Are the objectives of the study clearly articulated with a clear testable hypothesis stated?

-Is the study design appropriate to address the stated objectives?

-Is the population clearly described and appropriate for the hypothesis being tested?

-Is the sample size sufficient to ensure adequate power to address the hypothesis being tested?

-Were correct statistical analysis used to support conclusions?

-Are there concerns about ethical or regulatory requirements being met?

Reviewer #2: Prior comments were adequately addressed - please see below.

Reviewer #3: (No Response)

**Results**

-Does the analysis presented match the analysis plan?

-Are the results clearly and completely presented?

-Are the figures (Tables, Images) of sufficient quality for clarity?

Reviewer #2: Results are clearly and completely presented.

Reviewer #3: (No Response)

**Conclusions**

-Are the conclusions supported by the data presented?

-Are the limitations of analysis clearly described?

-Do the authors discuss how these data can be helpful to advance our understanding of the topic under study?

-Is public health relevance addressed?

Reviewer #2: Previous concerns regarding overstated conclusions have been sufficiently addressed.

Reviewer #3: (No Response)

**Editorial and Data Presentation Modifications?**

Reviewer #2: Please see final comments below.

Reviewer #3: (No Response)

**Summary and General Comments**

Reviewer #2: I appreciate the authors’ thoughtful responses to the reviewer comments and the substantial revisions made to the manuscript. The revised version is improved, and the authors have made a clear effort to address the concerns raised during the initial review.

In particular, the inclusion of validation using serum from LASV-infected animal models strengthens the study and provides an important step toward assessing assay performance in biologically relevant samples. The Discussion has also been revised to better reflect the limitations of the study and to avoid overstating the clinical applicability of the assay.

While the absence of validation using clinical samples from endemic settings remains a limitation, the authors now clearly acknowledge this and frame their work as a step toward the development of broadly reactive diagnostic tools rather than a fully validated diagnostic assay.

Overall, the manuscript is clearly written, the experimental approach is sufficiently sound, and the findings represent a useful contribution to the field.

Minor Comments:

1) While the serum samples from infected guinea pigs and mice are appropriately referenced to prior studies, it would be helpful to include a brief summary of key experimental details (e.g., virus strain, infection route, timing of collection) either in the Methods, Results, or table legend. This would allow readers to better interpret the findings without needing to consult the original references.

2) The use of infected animal sera is a valuable addition; however, these models may not fully recapitulate antigen dynamics observed in human infection. It would be helpful to ensure that these data are clearly framed as supportive of analytical performance rather than direct evidence of clinical diagnostic utility (lines 355-358).

3) The discussion of potential cross-reactivity with other arenaviruses is scientifically relevant; however, given the limited geographic overlap between LASV and New World arenaviruses, the practical diagnostic implications of this point may be more limited than implied (lines 433-438).

4) There are still minor issues with grammar and sentence structure throughout the manuscript. A final round of careful language editing would improve clarity and readability.

Reviewer #3: The authors have adequately addressed each of the reviewers comments.

PLOS authors have the option to publish the peer review history of their article (what does this mean?). If published, this will include your full peer review and any attached files.

Reviewer #2: No

Reviewer #3: **Yes:**Kathryn Hastie

**Figure resubmission:** While revising your submission, we strongly recommend that you use PLOS’s NAAS tool (https://ngplosjournals.pagemajik.ai/artanalysis) to test your figure files. NAAS can convert your figure files to the TIFF file type and meet basic requirements (such as print size, resolution), or provide you with a report on issues that do not meet our requirements and that NAAS cannot fix.  After uploading your figures to PLOS’s NAAS tool - https://ngplosjournals.pagemajik.ai/artanalysis, NAAS will process the files provided and display the results in the "Uploaded Files" section of the page as the processing is complete. If the uploaded figures meet our requirements (or NAAS is able to fix the files to meet our requirements), the figure will be marked as "fixed" above. If NAAS is unable to fix the files, a red "failed" label will appear above. When NAAS has confirmed that the figure files meet our requirements, please download the file via the download option, and include these NAAS processed figure files when submitting your revised manuscript.**Reproducibility:** To enhance the reproducibility of your results, we recommend that authors of applicable studies deposit laboratory protocols in protocols.io, where a protocol can be assigned its own identifier (DOI) such that it can be cited independently in the future. Additionally, PLOS ONE offers an option to publish peer-reviewed clinical study protocols. Read more information on sharing protocols at https://plos.org/protocols?utm_medium=editorial-email&utm_source=authorletters&utm_campaign=protocols

---

## [Editor Report · Decision Letter 2]

1 May 2026

Dear Dr. Maruyama,

We are pleased to inform you that your manuscript 'Development of novel monoclonal antibodies for detection of pan-Lassa virus' has been provisionally accepted for publication in PLOS Neglected Tropical Diseases.

Best regards,

Katharina Röltgen

Academic Editor

Michael Holbrook

Section Editor

Shaden Kamhawi

co-Editor-in-Chief

Paul Brindley

co-Editor-in-Chief

---

## [Editor Report · Acceptance letter]

Dear Dr. Maruyama,

We are delighted to inform you that your manuscript, "Development of novel monoclonal antibodies for detection of pan-Lassa virus," has been formally accepted for publication in PLOS Neglected Tropical Diseases.

Best regards,

Shaden Kamhawi

co-Editor-in-Chief

Paul Brindley

co-Editor-in-Chief
